# Effects of High-Dose Vitamin D Supplementation on Placental Vitamin D Metabolism and Neonatal Vitamin D Status

**DOI:** 10.3390/nu16132145

**Published:** 2024-07-05

**Authors:** Anna Louise Vestergaard, Matilde Kanstrup Andersen, Helena Hørdum Andersen, Krista Agathe Bossow, Pinar Bor, Agnete Larsen

**Affiliations:** 1Department of Obstetrics and Gynaecology, Randers Regional Hospital, 8930 Randers, Denmark; 2Department of Clinical Medicine, Aarhus University, 8200 Aarhus, Denmark; isipinbo@rm.dk; 3Department of Biomedicine, Aarhus University, 8000 Aarhus, Denmark; hha@biomed.au.dk (H.H.A.); al@biomed.au.dk (A.L.); 4Department of Obstetrics and Gynaecology, Aarhus University Hospital, 8200 Aarhus, Denmark

**Keywords:** neonates, obesity, placenta, vitamin D, vitamin D-binding protein, vitamin D metabolism, pregnancy

## Abstract

Vitamin D (vitD) deficiency (25-hydroxy-vitamin D < 50 nmol/L) is common in pregnancy and associated with an increased risk of adverse pregnancy outcomes. High-dose vitD supplementation is suggested to improve pregnancy health, but there is limited knowledge about the effects on placental vitD transport and metabolism and the vitD status of newborns. Comparing the current standard maternal supplementation, 10 µg/day to a 90 µg vitD supplement, we investigated placental gene expression, maternal vitD transport and neonatal vitD status. Biological material was obtained from pregnant women randomized to 10 µg or 90 µg vitD supplements from week 11–16 onwards. Possible associations between maternal exposure, neonatal vitD status and placental expression of the vitD receptor (*VDR*), the transporters (Cubilin, *CUBN* and Megalin, *LRP2*) and the vitD-activating and -degrading enzymes (*CYP24A1*, *CYP27B1*) were investigated. Maternal vitD-binding protein (VDBP) was determined before and after supplementation. Overall, 51% of neonates in the 10 µg vitD group were vitD-deficient in contrast to 11% in the 90 µg group. High-dose vitD supplementation did not significantly affect VDBP or placental gene expression. However, the descriptive analyses indicate that maternal obesity may lead to the differential expression of *CUBN*, *CYP24A1* and *CYP27B1* and a changed VDBP response. High-dose vitD improves neonatal vitD status without affecting placental vitD regulation.

## 1. Introduction

A sufficient vitamin D (vitD) supply during pregnancy is pivotal for both maternal, fetal and neonatal health. Over the years, observational studies have repeatedly linked maternal vitD deficiency to various adverse pregnancy outcomes like gestational diabetes, pre-eclampsia and fetal growth restriction [1,2,3,4,5]. In offspring, an insufficient vitD level in pregnancy is not only related to negative effects on bone development; numerous studies have also shown an increased risk of long-term health problems, such as diabetes, asthma, overweight and neurodevelopmental disorders [6,7,8,9].

The maternal blood stream is the sole source of vitD for the growing fetus. Maternal and placental factors are closely entangled and may affect fetal vitD supply. Within the placenta, active vitD, i.e., 1,25-dihydroxy-vitamin D (1,25(OH)_2_D), binds to the intracellular vitD receptor (VDR) [10], involved in the downstream regulation of numerous genes and physiological processes [11]. This presents a potential connection between maternal vitD status and placenta-associated pregnancy complications.

Placental uptake of vitD [12] is believed to mimic the active renal uptake of vitD [13,14], primarily facilitated by the megalin/cubilin complex, encoded by the *LRP2* and *CUBN* genes. The pro-hormone 25-hydroxy-vitamin D (25(OH)D) is transported in the blood stream, mostly bound to vitD-binding protein (VDBP) or, to a lesser extent, albumin [15]. Once reaching the placenta, VDBP-bound 25(OH)D is internalized by receptor-mediated endocytosis in the decidual cells and the syncytiotrofoblasts [12,16]. To reach the fetus, 25(OH)D crosses the placenta [16] and is activated by transformation into 1,25(OH)_2_D in the fetal kidney and in the fetal part of the placenta. Likewise, 25(OH)D may accumulate in the placenta or metabolize into either the active metabolite 1,25(OH)_2_D or the inactive metabolite 24,25(OH)_2_D through the interaction with CYP27B1 or CYP24A1, respectively, as both enzymes are present in the placental tissue [17,18,19,20].

In most countries, including Denmark, USA and UK, health authorities recommend a 10 µg/day (400 IU/day) vitD supplement during pregnancy [21,22]. Despite this recommendation, vitD deficiency (25(OH)D < 50 nmol/L) continues to be a global problem with a prevalence ranging from 46 to 87% [23,24] in various pregnant populations. Moreover, current recommendations often fail to consider maternal factors directly interfering with the supply of vitD to the fetus. Notably, obesity, a common condition in pregnancy today [25,26], is known to increase maternal vitD deficiency [27,28]. This is believed to be due to the sequestering of vitamin D in fat tissue, due to the fat-soluble property of this vitamin, resulting in a decreased amount of vitamin D in the bloodstream [29]. In our previous study, we found that placental *CYP24A1* gene expression increased with increasing maternal body weight [30]. Furthermore, it was previously reported that umbilical cord concentrations of 25(OH)D are affected by a high maternal body mass index (BMI) [31]. Though data are sparse, this underlines that a personalized approach may be needed to optimize nutrition in risk pregnancies such as those affected by maternal obesity [32]. The safety of markedly increasing the daily vitD supplementation up to 110 µg has been shown by others [33,34,35]. However, no previous supplementation study has focused on the impact of increased doses of maternal vitD supplementation on the placenta in a clinical setting and whether such placental factors could alter the amount of 25(OH)D reaching the fetus.

The aim of the present study was thus to investigate how increasing vitD supplementation to 90 µg/day during the last six months of pregnancy affects the umbilical cord blood concentration of 25(OH)D and placental vitD regulation, also considering if maternal pre-pregnancy BMI could be an independent factor affecting placental vitD metabolism.

## 2. Materials and Methods

### 2.1. Study Population

This study is a part of the randomized double-blinded trial GRAVITD registered at ClinicalTrial.gov on 17 February 2020 (NCT04291313). The study population and enrolment process are described in detail elsewhere [36]. In short, participants were invited during their first prenatal visit at the hospital (a free-of-charge nuchal translucency scan offered to all pregnant women in Denmark) in gestational week (GW) 11–16. Upon inclusion, participants were randomly allocated 1:1 to either 90 µg or 10 µg of vitD3 with even distribution over seasons. Exclusion criteria were age < 18 years, incapability to give written informed consent, disturbances in calcium metabolism, chronic kidney disease or a pre-existing high-dose (>10 µg/day) vitD supplementation initiated by a physician.

For all GRAVITD participants, serum samples were collected upon enrolment, i.e., GW 11–16. From approximately half of all participants, an additional sample was drawn in GW 24–38, at least three months apart from the first blood sample.

Of the 118 participants from whom placental samples were collected for this study (see Section 2.2), maternal blood samples from the first trimester (GW 11–16) were available from 113 pregnancies. For a smaller subgroup, blood samples from the third trimester (GW 29–38, n = 34) were available.

### 2.2. Placental Collection

Placental samples were obtained within the period of February 2021 to March 2022. Only placentas from singleton pregnancies not diagnosed with pre-eclampsia, gestational diabetes or intrauterine growth restriction were used for the placental analysis. Placentas were placed at 5 °C shortly after delivery. To be eligible, tissue had to be sampled within 5 h after delivery, in order to ensure sufficient tissue quality without interfering with clinical work at the maternity ward, leaving a total of 118 placental samples (Appendix A) [37] for this study. In short, a medial villous sample from a randomly selected slice of the placenta was collected, rinsed in isotonic saline, placed in RNAlater and stored at −20 °C until analysis.

### 2.3. Umbilical Cord Blood

Umbilical cord blood was drawn in EDTA tubes by a midwife and placed at 5 °C shortly after delivery. Plasma samples were eligible for this study if prepared and snap-frozen within 18 hours by the research team. All samples were kept at −80 °C until analysis. Not distinguishing between healthy and diseased pregnancies, a total of 472 umbilical cord blood samples were available for the present study.

### 2.4. 25(OH)D Analysis

The concentration of 25(OH)D in maternal-serum samples and cord blood plasma samples were determined at the Department of Clinical Biochemistry at Aarhus University Hospital, Denmark, using high-performance liquid chromatography coupled with tandem mass spectrometry, which is considered the golden standard of 25(OH)D assays [38].

### 2.5. VDBP Analysis

For a subgroup of participants (n = 101), the VDBP content of maternal blood was determined using a commercially available enzyme-linked immunosorbent assay kit (Cloud-Clone Corp, Houston, TX, USA) according to the manufacturer’s protocol. Samples were diluted 1:200,000. To avoid an undue influence from differences related to gestational age [39], only dyads (early/late pregnancy) of maternal blood samples that could fulfil strict criteria regarding sampling time were used, i.e., to be eligible for VDBP measures, the first-trimester sample must have been collected at GW 12 + 0–13 + 4, and the third-trimester sample had to be obtained at GW 34 + 0–35 + 6.

### 2.6. RNA Extraction

Total RNA was purified from 30 mg of homogenized villous placental tissue using an RNeasy Mini Kit (cat. no. 74104, Qiagen, Germantown, MD, USA) according to the manufacturer’s protocol. RNA concentrations were quantified by an absorbance measurement using a NanoDrop 2000 (Thermo Fisher Scientific, Waltham, MA, USA), and the quality of the extracted RNA was evaluated by agarose gel electrophoresis. cDNA was synthesized from 1000 ng of purified RNA using an ImProm™ Reverse Transcription System kit (Promega, Madison, WI, USA) with an oligoDt primer.

### 2.7. qPCR

The placental gene expression of vitD-related genes was assessed by qPCR specifically targeting genes encoding vitD transporters *CUBN* and *LRP2*, vitD-activating and -degrading enzymes *CYP27B1* and *CYP24A1* and the vitD receptor *VDR*.

For *CUBN*, *LRP2* and *VDR*, gene expression analyses were performed using Fast start SYBR green (cat. no. 4913914001, Roche, Basel, Switzerland) and the primers shown in Table 1 on a Light-Cycler 480 II and standardized to the geomean of *CycA* and *HPRT*. For *CYP24A1* and *CYP27B1*, gene expression analyses were performed using TaqMan Gene Expression (cat. no. 4369016, Applied Biosystems, Thermo Fisher Scientific, Waltham, MA, USA) and the probes shown in Table 1 on a QuantStudio 7 Flex (Applied biosystems) standardized to *HPRT*. Prior to statistical analysis, ΔCT (cycle threshold) values were converted into 2^(−ΔCT)^ values.

Only samples fulfilling all the technical requirements for evaluation were included, leaving 107 data points for the evaluation of *CUBN*, 114 data points for *LRP2* and 110 data points for *VDR*. In the *CYP27B1* qPCR, the gene expression was below the detection limit in 46 samples. For these samples, the mean CT value of *CYP27B1* was set to be 37 in the statistical analysis as this was the detection limit.

### 2.8. Statistics and Descriptive Analysis

All statistical analyses comparing the vitD dosing groups were performed as an intention to treat based on the randomization to 10 µg or 90 µg vitD supplementation. The dosing groups were compared using a Student’s *t*-test for the assessment of continuous variables when comparing two groups and a one-way ANOVA when comparing more than two groups. A chi-squared test was used for categorical variables. Continuous data were analyzed for a Gaussian distribution using QQ-plots, and a logarithmic transformation of the data was performed when appropriate.

In addition to the analysis based on the dosing groups, we investigated the possible associations with maternal 25(OH)D concentration using linear correlation analyses. Potential homoscedasticity was evaluated by the visual inspection of residual plots, and the normality of residuals was considered by the visual inspection of residual QQ plots.

We further described how women with pre-pregnancy overweight or obesity reacted to the vitD supplementation, using a spline model approach, stratifying according to the vitD dosing group. Spline models were constructed based on four not-prespecified knots in each model. The spline models were used for descriptive analysis only in order to visualize any potential association between concentration or expression and BMI in the two dosing groups. Spline models serve as a complement to categorical analyses and linear regression models and give a visual of the association within different intervals accommodating the complex nature of human biology. They are piecewise polynomials defined by the knots and smoothened to create a curve [40].

All analyses were performed using STATA (version 18, StataCorp, College Station, TX, USA) and Prism (version 10.2.2, GraphPad Software, Boston, MA, USA). A *p*-value < 0.05 was considered statistically significant.

## 3. Results

### 3.1. Placental Expression of vitD-Related Genes

Prior to comparing the placental gene expression of the two dosing groups, we investigated the pregnancy characteristics of each group, finding no differences in the distribution of offspring sex or overall maternal demographics, except for a slight difference in mean gestational age at delivery (10 µg: 40.2 weeks vs. 90 µg: 39.8 weeks, *p* = 0.031) (Appendix A).

Investigating the gene expression of the vitD-related genes *CUBN*, *LRP2*, *CYP24A1*, *CYP27B1* and *VDR*, we found no statistically significant difference between the two vitD dosing groups (Figure 1a–e).

#### 3.1.1. Correlation to the Third-Trimester Maternal Serum 25(OH)D

In a subset of the participants, it was possible to analyze gene expression in relation to maternal 25(OH)D concentration in the third trimester (n = 34). A linear regression model revealed that maternal third-trimester 25(OH)D concentrations were positively correlated with the expression of *LRP2* (*p* = 0.036), but not *CUBN* (*p* = 0.44), *CYP24A1* (*p* = 0.42), *CYP27B1* (*p* = 0.82) or *VDR* (*p* = 0.074) (Figure 2a–e).

#### 3.1.2. The Effect of Pre-Pregnancy BMI

As depicted in Figure 3, no difference was found in the gene expression of any of the vitD-related genes among women with a pre-pregnancy BMI < 30 and >30 kg/m^2^. To further nuance the possible effect of BMI, we used a spline model to visualize any tendency toward an effect of pre-pregnancy BMI. This approach showed a tendency of an increased *CUBN* expression with increasing BMI in the 90 µg vitD group (Figure 4a). No differences with regard to *LRP2* expression were observed (Figure 4b). For *CYP24A1*, the spline model approach indicated a higher *CYP24A1* expression among overweight or obese women in the 90 µg vitD group with no apparent response to increasing BMI in the 10 µg vitD group (Figure 4c). With regard to *CYP27B1*, the 90 µg vitD group appeared to have a lower expression of *CYP27B1* compared to the 10 µg vitD group in normal and overweight pregnancies, but at a BMI of 29 kg/m^2^, this correlation was reversed, and *CYP27B1* expression was higher in the 90 µg vitD group (Figure 4d). For *VDR*, the spline model did not point toward an association between pre-pregnancy BMI and placental *VDR* expression (Figure 4e).

#### 3.1.3. Impact of Early vitD Status

As the randomization was conducted at the end of the first trimester, weeks after the placentation, it is relevant to test if the maternal 25(OH)D concentration at study entry had any association to the placental gene expression at delivery. Using a linear correlation model, we found no association between first-trimester 25(OH)D concentration and the gene expression of *CUBN* (*p* = 0.68), *LRP2* (*p* = 0.30), *CYP24A1* (*p* = 0.39) or *CYP27B1* (*p* = 0.87). However, for the gene expression of *VDR*, a statistically significant positive correlation (*p* = 0.02) was found (Figure 5).

#### 3.1.4. Delivery Mode and Offspring Sex

As we have previously found that fetal sex is reflected in the placental transcriptome [41], we tested if the vitD-associated genes were differently expressed in placentas from pregnancies with a male offspring compared to their female counterparts, finding no statistically significant impact of offspring sex for any of the analyzed genes (*CUBN* (*p* = 0.39), *LRP2* (*p* = 0.85), *CYP24A1* (*p* = 0.92), *CYP27B1* (*p* = 0.93) or *VDR* (*p* = 0.13)). We also tested if mode of delivery (vaginal vs. sectio) altered the gene expression but found no association for any of the analyzed genes (*CUBN* (*p* = 0.81), *LRP2* (*p* = 0.35), *CYP24A1* (*p* = 0.45), *CYP27B1* (*p* = 0.75) or *VDR* (*p* = 0.21)).

### 3.2. Maternal VDBP during Pregnancy

Overall, offspring sex and maternal demographic data were similar in the two dosing groups among the subset of participants (n = 101) in whom VDBP was measured in maternal serum, albeit the 90 µg vitD group did include more women of non-Scandinavian origin, although the vast majority of women were Scandinavian in both groups (Appendix A). VDBP results were available from 99 participants.

As VDBP was determined in GW 12 + 0–13 + 4 and GW 34 + 0–35 + 6, respectively, we initially correlated the absolute values to maternal 25(OH)D concentration at these time points, finding no linear correlation between maternal 25(OH)D and VDBP in neither the first (*p* = 0.14) nor the third trimester (*p* = 0.64) (Figure 6). Moreover, the VDBP concentration in neither the first nor third trimester was associated with offspring sex (*p* = 0.65 and *p* = 0.72, respectively).

The mean VDBP concentration measured in the third trimester was lower in the 90 µg group, i.e., 46,496 ng/mL, 95% CI [38,716, 54,275], compared to 56,385 ng/mL, 95% CI [48,230, 64,541] in the 10 µg group; however, this difference was not statistically significant (*p* = 0.08).

When calculating the absolute and relative VDBP change (first vs. third trimester), we found no statistically significant difference between the two dosing groups (*p* = 0.41 and *p* = 0.94, respectively). However, when we investigated the possible effect of pre-pregnancy BMI using spline models for the absolute (Figure 7a) and relative VDBP change (Figure 7b), we observed that the relative change in VDBP appeared to be affected by increasing BMI in the 90 µg group, whereas BMI did not elicit a marked VDBP change in the 10 µg dosing group.

### 3.3. Neonatal vitD Status

As the fetal vitD supply is based on the transplacental vitD transfer, we finally determined the vitD status in the umbilical cord blood from 472 newborns. The offspring sex and maternal demographics of this subset of the GRAVITD cohort, which did not distinguish between healthy and complicated pregnancies, did not find any significant differences in the demographic and clinical factors between the two dosing groups (Appendix A).

It was evident that newborns from the 90 µg dosing group had a markedly higher mean 25(OH)D concentration compared to the 10 µg group, i.e., 83.5 nmol/L, 95% CI [80.0, 87.1] vs. 50.9 nmol/L, 95% CI [48.4, 53.4], (*p* < 0.0001).

Correspondingly, the prevalence of vitD deficiency was markedly reduced among newborns from the high-dose vitD group, i.e., 11 % (n = 26) in contrast to a prevalence of 51% (n = 124) in the 10 µg group. A total of 6% (n = 14) of newborns in the 10 µg vitD group had severe vitD deficiency <25 nmol/L, while this was only the case for 3 % (n = 7) of newborns in the 90 µg group.

Looking at the offspring from women with a pre-pregnancy BMI > 30 kg/m^2^ specifically, we found an increased frequency of vitD deficiency in the offspring from these pregnancies in both dosing groups, i.e., 17% in the 90 µg group and 61% in the 10 µg group.

## 4. Discussion

The findings from this study support that increased maternal vitD supplementation significantly reduces vitD deficiency among newborns and has a beneficial effect on neonatal vitD status, without signs of a reduced placental vitD uptake or an increased placental vitD degradation. The effects of high-dosage vitD supplementation on placental function have not previously been scrutinized. In this study, we analyzed a large number of placentas from pregnant women supplemented with either 10 µg or 90 µg of vitD3 aiming at reducing this knowledge gap.

Although our analysis based on an intention to treat found no significant differences between the dosing groups, there were indications that high maternal 25(OH)D concentrations may lead to an increased transplacental transport capacity of vitD. Though limited by the fact that serum collected later in the pregnancy, e.g., in the third trimester was only available from around one third of the participants, we thus found that an increased maternal third-trimester 25(OH)D concentration was associated with increased placental *LRP2* expression. This is supported by a previous small-scale study by Park et al. who further reported a positive correlation with placental *CUBN* [19]. Previous studies are, however, not conclusive on this point as others did not find any correlation between placental *CUBN* and *LRP2* expression and maternal 25(OH)D in the third trimester [12]. In the present study, we investigated the effect of two markedly different vitD supplementation doses on placental gene expression which was not conducted in the previous studies [12,19]. Our findings thus resemble the results one would see if clinical recommendations were altered. However, there are some limitations such as the absence of pill counts, and the set-up did not make it possible to register individual variations in sun exposure. The RCT set-up considers seasonal variation during randomization, but it is possible that the increased vitD uptake would be more profound if data from the winter season were analyzed separately. Also, our analysis might reflect that it was carried out in a cohort with a limited frequency of vitD deficiency, especially very limited numbers of severe vitD deficiency, just as complicated pregnancies were not included in the placental study.

As previous research has shown that 25(OH)D bound to VDBP rather than free 25(OH)D is preferentially taken up by the placenta through the megalin/cubilin complex [12,19,42], it could be hypothesized that alterations in VDBP concentration may also influence placental vitD uptake. However, we did not observe a significant increase in VDBP associated with a difference in vitD supplementation. We cannot exclude that the sample size in which VDBP was tested and the fact that we are employing an intention-to-treat analysis may affect the results. On the other hand, a natural rise in VDBP of 40–50% in pregnancy has been described with a peak in the early third trimester [15,39], indicating that in the pregnant state, vitD metabolism is already maximized to adequately supply the fetus.

Interestingly, the spline model did hint toward an effect on the change in VDBP over time related to pre-pregnancy BMI. This could indicate that increasing the vitD supplement of women with high BMI to 90µg triggers an increase in VDBP that allows them to make more use of the vitD given. Similarly, our spline model also indicated an increase in *CUBN* in this group of women. On the other hand, the 10 µg of vitD supplement did not appear to induce a change in VDBP or *CUBN*, even though these women might need more vitD.

In a previous study, we found a positive association between the gene expression of *CYP24A1* in term placentas and pre-pregnancy BMI [30]. Though not statistically significant in this study, our spline model did indicate an increased expression of *CYP24A1* in obese and overweight pregnancies. This could support a higher vitD need in obese pregnancies, as the frequency of vitD-deficient newborns was higher in these pregnancies. However, further studies are warranted to examine the maternal, placental and fetal responses to high-dose vitD supplementation in pregnancy, further comparing obese, overweight and lean women to determine if the advice on vitD supplementation should consider body composition.

With regard to our study population as a whole, this study shows that the currently recommended vitD supplementation of 10 µg/day does not prevent vitD deficiency in as many as 51% of all newborns, whereas increasing the supplementation to 90 µg results in vitD sufficiency in the majority of newborns. Moreover, the prevalence of vitD deficiency might even be underestimated in the present study, as 25(OH)D concentrations determined in plasma, as was the case with the umbilical cord samples, tend to be higher than those measured in serum [43].

From a health perspective, our findings are of high clinical relevance as a sufficient fetal and neonatal vitD status is important for bone health [44,45]. Furthermore, intrauterine vitD exposure may potentially reduce later-life disease risk by strengthening the immune system [46], thereby potentially reducing the risk of immune-system-related conditions like type 1 diabetes, asthma and allergy [6,9,47]. VitD insufficiency during pregnancy and the first vulnerable years of life also markedly increases the risk of common and potentially fatal infections such as respiratory syncytial virus (RSV) [48]. A previous study found a six-fold increase in RSV frequency in children who were vitD-deficient at birth [49], and a reduction in RSV susceptibility would reduce hospitalization rates and mortality in young children.

Placental *VDR* expression per se has previously been linked to fetal growth and development [12]. Though we found no direct association between supplementation or actual maternal vitD status in late pregnancy and *VDR*, our previous next-generation RNA sequencing study on vitD supplementation and the placental transcriptome found that alterations in placental function occur with the high-dose regime tested here [41]. Such changes, e.g., those linked to neurodevelopment, are likely to further benefit the growing fetus, in addition to the direct increase in fetal vitD supply. Notably, we found that *VDR* expression appeared unaffected by high-dose supplementation through the last six months of pregnancy, albeit significantly associated with first-trimester vitD measurements, highlighting the importance of also paying specific attention to pre-pregnancy vitD status for optimal placental function.

As VDR activation depends on the presence of the active vitD metabolite 1,25(OH)_2_D, we further investigated the effects of vitD supplementation on the gene encoding the vitD-activating enzyme *CYP27B1*. Notably, we found that the placental gene expression of *CYP27B1* was below the detection limit in 39% of the samples. This likely relates to the low content of the enzyme in villus tissue compared to other parts of the placenta [42]. We did not see significant effects of either dosing group or maternal vitD concentration. However, we did observe indications of differences in *CYP27B1* expression in relation to pre-pregnancy BMI. This could further suggest a complex vitD metabolism and uptake in women with obesity and overweight that may be associated with the numerous clinical problems seen in such pregnancies; however, more research is needed before such conclusions can be made for certain.

## 5. Conclusions

A high dose of vitD supplementation during pregnancy significantly reduces vitD deficiency among newborns and thus improves the vitD status without remarkably reducing the placental vitD uptake or affecting placental vitD metabolism. There is, however, a need for further studies on vitD metabolism and maternal–fetal vitD transfer in women with overweight or obesity to elucidate if maternal body weight affects the nutritional needs in pregnancy.

## Figures and Tables

**Figure 1 nutrients-16-02145-f001:**
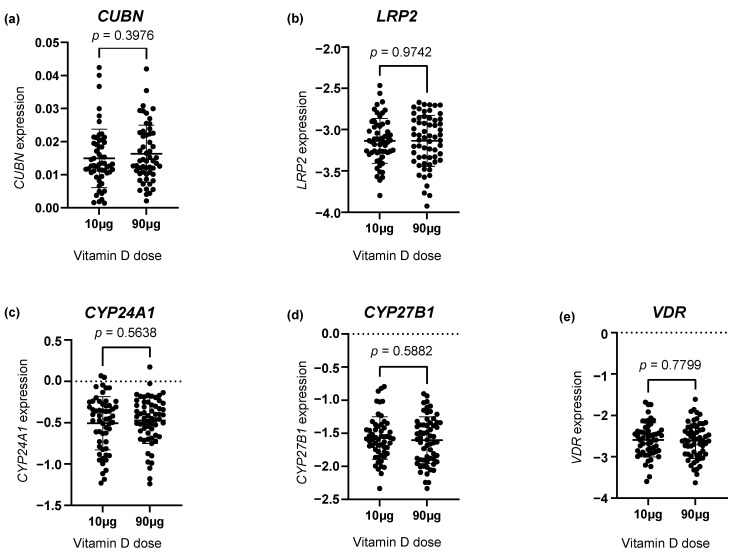
The placental gene expression of (**a**) *CUBN*, (**b**) *LRP2*, (**c**) *CYP24A1*, (**d**) *CYP27B1* and (**e**) *VDR* in the two vitamin D dosing groups (10 µg vs. 90 µg). The error bars in the figure illustrate the mean expression ± SD. *p*-values were calculated using a Student’s *t*-test. A logarithmic transformation of *LRP2*, *CYP24A1*, *CYP27B1* and *VDR* expression values was performed to achieve a Gaussian distribution of the data. Note that *CYP27B1* expression was under the detection limit in 46 placental samples of which 28 were from the 90 µg vitamin D group and 18 from the 10 µg vitamin D group. *CUBN*, *LRP2* and *VDR* expression was standardized to the geomean of *CycA* and *HPRT*, and *CYP24A1* and *CYP27B1* expression was standardized to the expression of *HPRT*.

**Figure 2 nutrients-16-02145-f002:**
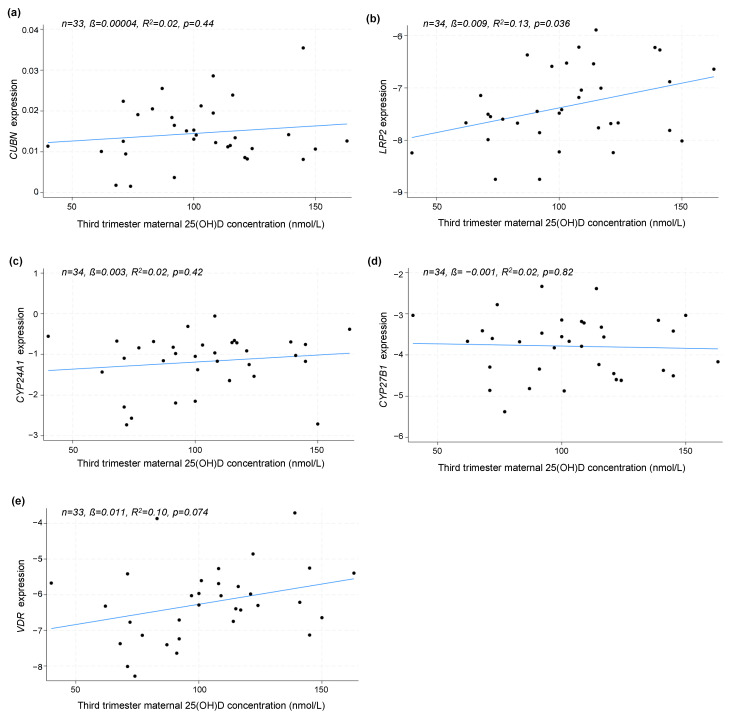
The linear correlation analysis between the placental gene expression of (**a**) *CUBN*, (**b**) *LRP2*, *(***c**) *CYP24A1*, (**d**) *CYP27B1* and (**e**) *VDR* and maternal 25(OH)D concentration in the third trimester. The correlation is solely based on the measured 25(OH)D in the third trimester, not distinguishing between dosing groups. A logarithmic transformation of *LRP2*, *CYP24A1*, *CYP27B1* and *VDR* expression values was performed to achieve a Gaussian distribution of the data. *CUBN*, *LRP2* and *VDR* expression was standardized to the geomean of *CycA* and *HPRT*, and *CYP24A1* and *CYP27B1* expression was standardized to the expression of *HPRT*. 25(OH)D, 25-dihydroxy vitamin D.

**Figure 3 nutrients-16-02145-f003:**
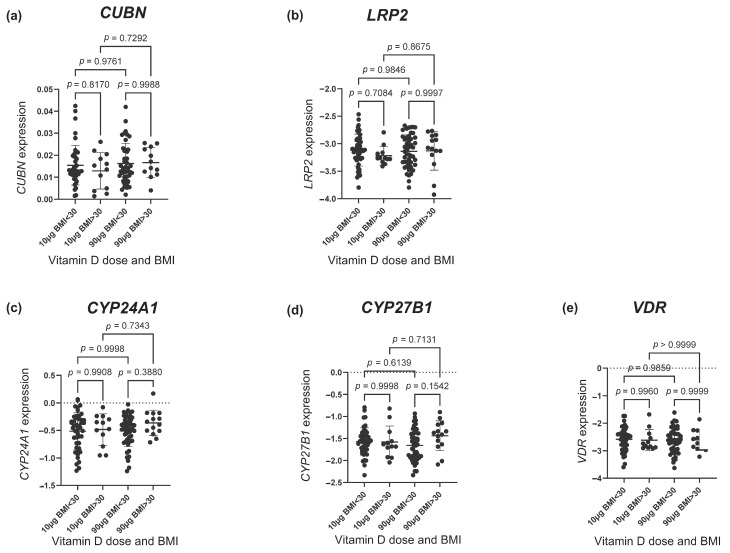
The gene expression of (**a**) *CUBN*, (**b**) *LRP2*, (**c**) *CYP24A1*, (**d**) *CYP27B1* and (**e**) *VDR* according to the vitamin D dosing group (10 µg vs. 90 µg) and BMI group (> and <30 kg/m^2^). The error bars in the figure illustrate the mean expression ± SD. *p*-values were calculated using a one-way ANOVA. A logarithmic transformation of *LRP2*, *CYP24A1*, *CYP27B1* and *VDR* expression values was performed to achieve a Gaussian distribution of the data. Note that *CYP27B1* expression was under the detection limit in 46 placental samples of which 28 were from the 90 µg vitamin D group and 18 from the 10 µg vitamin D group. *CUBN*, *LRP2* and *VDR* expression was standardized to the geomean of *CycA* and *HPRT*, and *CYP24A1* and *CYP27B1* expression was standardized to the expression of *HPRT*. BMI, body mass index.

**Figure 4 nutrients-16-02145-f004:**
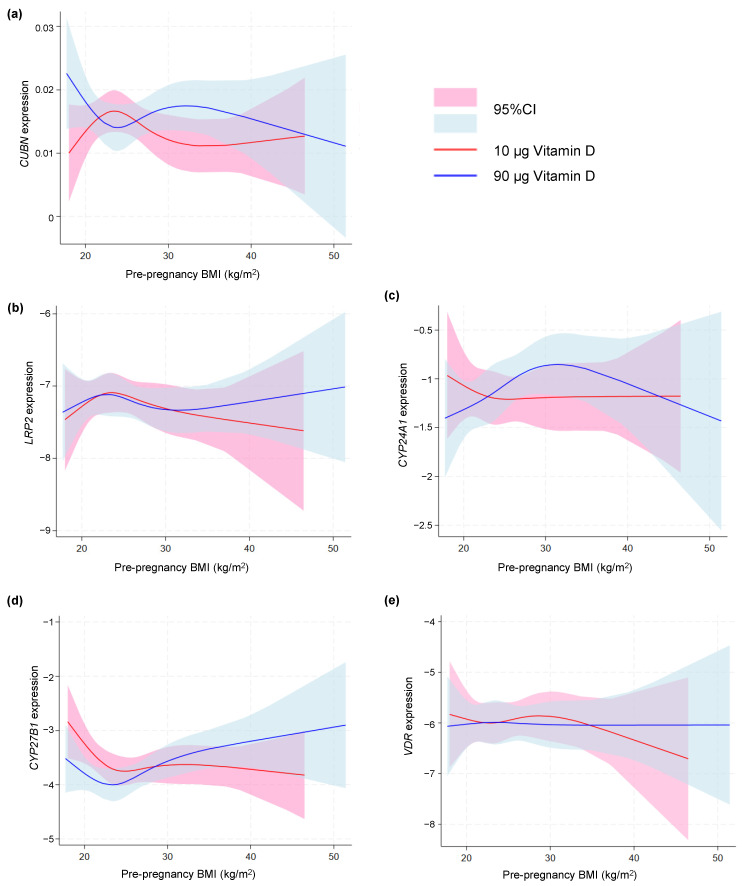
Spline models depicting the gene expression of (**a**) *CUBN*, (**b**) *LRP2*, (**c**) *CYP24A1*, (**d**) *CYP27B1* and (**e**) *VDR* correlated to maternal pre-pregnancy BMI in the two vitamin D dosing groups (10 µg vs. 90 µg). A logarithmic transformation of *LRP2*, *CYP24A1*, *CYP27B1* and *VDR* expression values was performed to achieve a Gaussian distribution of the data. Note that *CYP27B1* expression was under the detection limit in 46 placental samples of which 28 were from the 90 µg vitamin D group and 18 from the 10 µg vitamin D group. *CUBN*, *LRP2* and *VDR* expression was standardized to the geomean of *CycA* and *HPRT*, and *CYP24A1* and *CYP27B1* expression was standardized to the expression of *HPRT*. BMI, body mass index; 95% CI, 95% confidence interval.

**Figure 5 nutrients-16-02145-f005:**
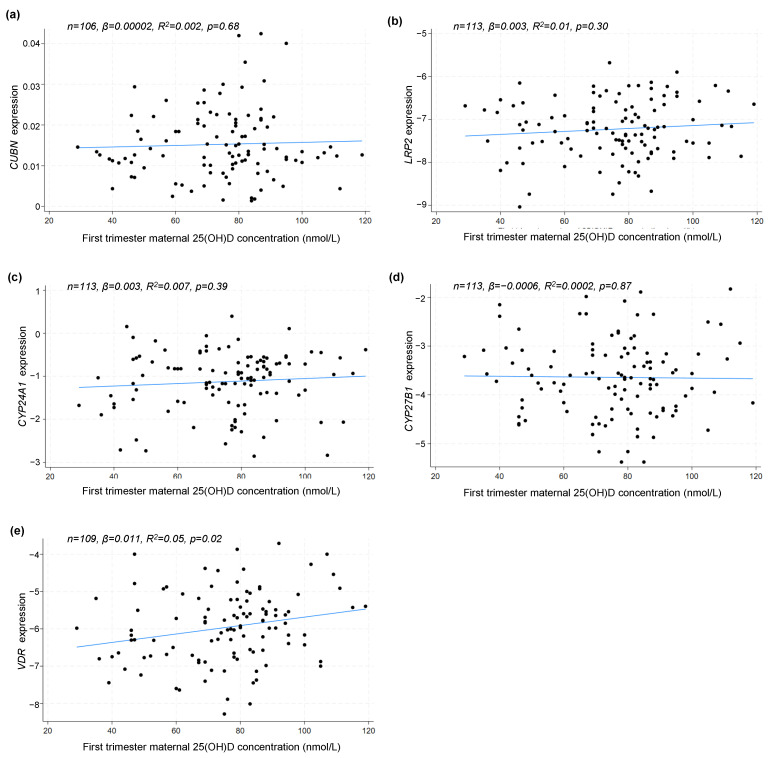
Linear correlation analyses between placental (**a**) *CUBN*, (**b**) *LRP2*, (**c**) *CYP24A1*, (**d**) *CYP27B1* and (**e**) *VDR* expression and maternal 25(OH)D concentration in the first trimester. A logarithmic transformation of *LRP2*, *CYP24A1*, *CYP27B1* and *VDR* expression values was performed to achieve a Gaussian distribution of the data. *CUBN*, *LRP2* and *VDR* expression was standardized to the geomean of *CycA* and *HPRT*, and *CYP24A1* and *CYP27B1* expression was standardized to the expression of *HPRT*. 25(OH)D, 25-hydroxy vitamin D.

**Figure 6 nutrients-16-02145-f006:**
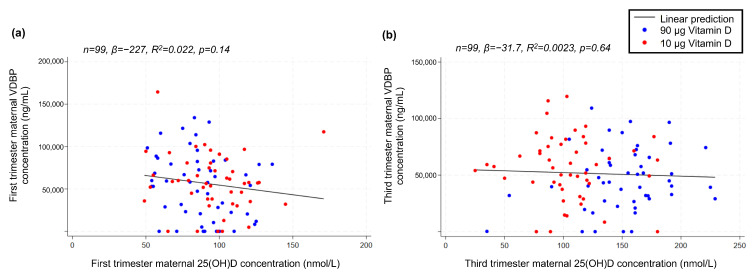
Linear correlations between (**a**) maternal-serum VDBP and 25(OH)D concentration in the first trimester and (**b**) maternal-serum VDBP and 25(OH)D concentration in the third trimester. VDBP, vitamin D-binding protein; 25(OH)D, 25-hydroxy vitamin D.

**Figure 7 nutrients-16-02145-f007:**
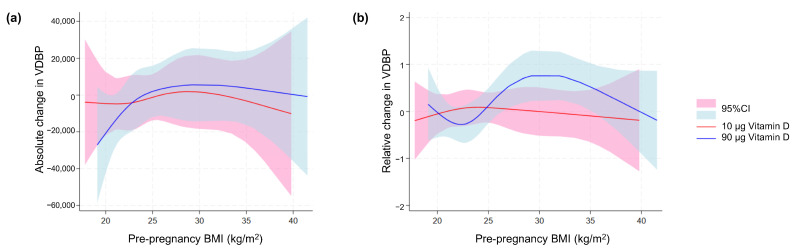
VDBP in maternal blood in relation to maternal body weight. Spline models of the association between maternal pre-pregnancy BMI and (**a**) the absolute change and (**b**) relative change in maternal-serum VDBP concentration from the first to third trimester in relation to and stratified for the vitamin D dosing group (10 µg vs. 90 µg). BMI, body mass index; VDBP, vitamin D-binding protein; 95% CI, 95% confidence interval.

**Table 1 nutrients-16-02145-t001:** Primers and probes used for qPCR.

Gene	Primer
*CUBN*	
Forward	5′-CTGGACGGCCATTACTCACA-3′
Reverse	5′-CTGAGACTGGAAGACGGCAG-3′
*LRP2*	
Forward	5′-GCCCTTTCGCTGTCCTAGTT-3′
Reverse	5′-AGGGCTCTTGAACACACTCG-3′
*VDR*	
Forward	5′-GCCTGACCCTGGAGACTTTG-3′
Reverse	5′-GGGCAGGTGAATAGTGCCTT-3′
*HPR1T*	
Forward	5′-CCTGGCGTCGTGATTAGTGA-3′
Reverse	5′-GAGCACACAGAGGGCTACAA-3′
*CycA*	
Forward	5′-GCCGAGGAAAACCGTGTACTA-3′
Reverse	5′-ACCCTGACACATAAACCCTGG-3′
	**Probe**
*CYP24A1*	HS00167999_m1
*CYP27B1*	HS01096154_m1
*HPRT1*	HS02800695_m1

## Data Availability

The full dataset is not publicly available as we do not have consent from the participants to publish the full dataset. However, a de-identified dataset will be available from the corresponding author upon reasonable request.

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
