# Peer review of "Effects of High-Dose Vitamin D Supplementation on Placental Vitamin D Metabolism and Neonatal Vitamin D Status"

_nutrients, 2024, doi:10.3390/nu16132145_

Round 1

Reviewer 1 Report

Comments and Suggestions for Authors

I have read this paper with great interest with a background of clinical research in perinatal setting. In essence, it reports on a secondary, exploratory analysis as substudy to a RCT on normal versus high dose vitamin D suppletion during pregnancy (cf to illustrate, figure 5a-e), assessing maternal, umbilical cord and placental outcome variables within this study design.

I value the paper, but it should be much clearer that this secondary study was not powered, so that any positive or negative final is rather an association, not necessary causal. The rct and the limitations should likely be further stressed in the abstract.

My second concern related to the ‘biomakers’ used. Technically, RNA and not protein or protein activity has been measured. This is also a limitation, as (likely even more relevant for transporters), RNA expression does not necessary equal protein activity.

What is the mechanistic link between obesity and vitamin D plasma levels ? secondary to nutritional intake, higher distribution volume related to higher fat mass, or inflammation ?

Minor, editing suggestions 

Fetal, line 33: suggest to add neonatal

Line 43-44: please check sentence on language

Line 199, please check ‘participant’

Line 212: were no difference

Line 215: Tendency to ? that’s quite ‘poor’ language as indeed not significant

Author Response

We thank the reviewers for taking the time to review our manuscript and value their through and constructive remarks on the paper

Comment 1: I have read this paper with great interest with a background of clinical research in perinatal setting. In essence, it reports on a secondary, exploratory analysis as substudy to a RCT on normal versus high dose vitamin D suppletion during pregnancy (cf to illustrate, figure 5a-e), assessing maternal, umbilical cord and placental outcome variables within this study design. 

I value the paper, but it should be much clearer that this secondary study was not powered, so that any positive or negative final is rather an association, not necessary causal. The rct and the limitations should likely be further stressed in the abstract. 

Response 1: The reviewer raises some concern regarding the power of the study. We would like to clarify that the number of total participants in the GRAVITD trial is powered based on the clinical outcomes (frequency of pre-eclampsia and gestational diabetes). We would therefor like to stress that it was never the intention to conduct placentas from all participants in the GRAVITD trial, and based on the literature and our own previous experience within our research team (Justesen et al, Sci Rep, 2023) including a NGS study of the placental transcriptome (Vestergaard et al, Nutrients, 2023) we know that we can identify biological changes between two different groups when we have around 70 samples. The present study was based on placental samples from 118 uncomplicated pregnancies collected between January 2021 and January 2022, a collection period chosen to ensure in order to ensure that the study takes seasonal variation into account. A sample size of this proportion in a molecular biological study is considered a large and sufficient number of samples to identify differences between the two exposures 10 vs 90µg vitamin D based on the 1:1 randomization at inclusion. This is also reflected by the fact that previous studies from other research teams in the field have been based on smaller sample sizes and have still been able to detect changes (Park et al, The American journal of clinical nutrition., 2017. Ashley et al, Elife, 2022.) So even though we certainly agree with the reviewer that it is of great value to obtain a high number of biological samples for this type of study, we respectively but firmly believe that in respect to the hypothesis tested, power is not a concern of this molecular study.

Comment 2: My second concern related to the ‘biomakers’ used. Technically, RNA and not protein or protein activity has been measured. This is also a limitation, as (likely even more relevant for transporters), RNA expression does not necessary equal protein activity. 

Response 2: The reviewer rightly noticed that we have used mRNA expression as a tool to reflect changes in placental function in the present study. and not protein expression or protein activity. We did this as our lab has great experience with this kind of analysis for purposes similar to the aim of the present study, a mode of study also commonly used in the field. The benefit of genetic expression being that within the practical circumstances i.e. collection of tissue from a large number of births regardless of labour method (vaginal or C-section), we have the best chance to obtained a large number of samples of a good quality for analysis without interfering with clinical procedures, e.g. in contrast to protein activity, which would limit our time frame and likely make the use of material from vaginal births almost impossible. The advantages of analysis of mRNA expression also being that it gives us the opportunity to analyze a number of targets in the same tissue samples from a large number of placentas as was the aim of the present study. Of course, we cannot completely exclude that post-translations changes in protein expression or activity could additionally affect placental function, however, given that the exposure of vitamin D in the dose of 10 µg or 90 µg is given for 6 months the dose would most likely have altered the mRNA expression well before tissue collection was performed. This further supports that mRNA is a likely biomarker for vitamin D induced changes in this context, even more so, also keeping in mind that the vitamin D receptor (VDR) works primarily as an intracellular receptor leading to downstream regulation of numerous genes through repression or activation of gene transcription. While we agree with the reviewer that protein quantification would also have benefits, such methods would not have been without limitations as a Western blot would depend on the quality of the used antibody. Moreover, the degradation of protein within the biological tissue would appear more quickly after delivery of the placenta and due to our clinical setup again increasing the risk of interfering with the work of the midwifes or losing to many samples for our study. Thus, we found the best method for our study was quatification of mRNA.

Comment 3: What is the mechanistic link between obesity and vitamin D plasma levels ? secondary to nutritional intake, higher distribution volume related to higher fat mass, or inflammation ? 

Response 3: We agree with the reviewer that the text would benefit from touching upon this subject and for clarification have expanded the introduction accordingly. Please see line 63-65.

Comment 4: Minor, editing suggestions 

Response 4: We thank the reviewer for the thorough revision and have seen to the suggestions presented below.

Comment 4a: Fetal, line 33: suggest to add neonatal

Response 4a: Thank you for the good suggestion, this has now been added in line 33

Comment 4b: Line 43-44: please check sentence on language

Response 4b: This sentences has now been changed as requested, see line 43

Comment 4c: Line 199, please check ‘participant’

Response 4c: This has now been changed to participants.

Comment 4d: Line 212: were no difference

Response 4d: The sentence has now been corrected, see line 216

Comment 4e: Line 215: Tendency to ? that’s quite ‘poor’ language as indeed not significant

Response 4e: There was a technical error in the naming of the figures resulting in a false figure referral in line 215, this has now been corrected. We hope that by correcting this it has become clearer what we are referring to. We also added some extra information about the spline model approach in the methods section (se line 178-181) which could help clarify the kind of change we are referring to in line 220 (was 215 before).

Reviewer 2 Report

Comments and Suggestions for Authors

In the manuscript submitted to me for review entitled "Effects of High-dose Vitamin D Supplementation on Placental Vitamin D Metabolism and Neonatal Vitamin D Status the authors Anna Louise Vestergaard, Matilde Kanstrup Andersen, Helena Hørdum Andersen, Krista Agathe Bossow, Pinar Bor and Agnete Larsen present a study in which pregnant women were given supplements containing 10 µg/day or 90 µg/day vitD. The authors investigated placental gene expression, maternal vitD transport and neonatal vitD status in the observed patients and their newborn infants.

The conducted study presents extremely important information proving that the use of the higher concentration of vitD leads to a decrease in vitD deficiency in newborns, which could contribute to the normal prescription of higher doses of vitD during pregnancy in the future. The study also points to the need to further investigate whether pre-pregnancy obesity affects vitD metabolism and vitD transfer from mother to fetus.

The study included pregnant women from 11-16 weeks of gestation onwards. The study was conducted in accordance with the Declaration of Helsinki and approved by the scientific ethical committee of Central Denmark Region. Prior consent was obtained from each patient for their inclusion in the study.

To support their research, the authors used 47 references that present information from studies published over the past two decades. Nearly 1/3 of the total references are from the last 5 years, indicating that the topic of the human health benefits of vitD intake is current and would be of interest to Nutrients readers. I did not notice any redundant self-citations, all references used are appropriate and necessary for the preparation of the manuscript.

My remarks and recommendations to the authors are:

1. I think it would be useful for readers if the Spline model approach is described in a little more detail.

2. A technical spelling error was made - the caption is missing Figure 3. Figure 4 is indicated twice. Let the correct numbering be inserted.

3. In the supplementary file, very important information from the study is given. I personally think that if Tables S2, 3 and 4 were in the main text of the manuscript it would contribute to a better understanding of the obtained results. This is my opinion, of course the authors decide which information to leave in the manuscript and which in the supplementary file.

Author Response

We thank the reviewer for taking the time to review our manuscript and value the through and constructive remarks on the paper

Comment 1: I think it would be useful for readers if the Spline model approach is described in a little more detail.

Response 1: We thank the reviewer for pointing out that it would be useful to the readers if this model is described in more details and have edited the text accordingly. Please see line 178-181.

Comment 2: A technical spelling error was made - the caption is missing Figure 3. Figure 4 is indicated twice. Let the correct numbering be inserted.

Response 2: We thank the reviewer for letting us know about the technical spelling error. Figure 3 (the one with gene expression and BMI) has now been named correctly.

Comment 3: In the supplementary file, very important information from the study is given. I personally think that if Tables S2, 3 and 4 were in the main text of the manuscript it would contribute to a better understanding of the obtained results. This is my opinion, of course the authors decide which information to leave in the manuscript and which in the supplementary file.

Response 3: We appreciate the reviewers view on this matter. Some tables have been made supplementary in the hope to obtain a better clarity of the manuscript. However, we do agree that they would also be valuable to have within the manuscript. Therefore, we will let the editor and reviewers decide if the tables S2, S3 and S4 should remain as supplementary tables or if they should be placed within the main text.

Round 2

Reviewer 1 Report

Comments and Suggestions for Authors

no additional comments